# Preclinical Characterization of the ^177^Lu-Labeled Prostate Stem Cell Antigen (PSCA)-Specific Monoclonal Antibody 7F5

**DOI:** 10.3390/ijms24119420

**Published:** 2023-05-29

**Authors:** Franziska Striese, Christin Neuber, Sandy Gräßel, Claudia Arndt, Martin Ullrich, Jörg Steinbach, Jens Pietzsch, Ralf Bergmann, Hans-Jürgen Pietzsch, Wiebke Sihver, Marcus Frenz, Anja Feldmann, Michael P. Bachmann

**Affiliations:** 1Helmholtz-Zentrum Dresden-Rossendorf, Institute of Radiopharmaceutical Cancer Research, 01328 Dresden, Germany; f.striese@hzdr.de (F.S.); c.neuber@hzdr.de (C.N.); s.weissflog@hzdr.de (S.G.); c.arndt@hzdr.de (C.A.); m.ullrich@hzdr.de (M.U.); j.steinbach@hzdr.de (J.S.); j.pietzsch@hzdr.de (J.P.); r.bergmann@hzdr.de (R.B.); h.j.pietzsch@hzdr.de (H.-J.P.); w.sihver@hzdr.de (W.S.); a.feldmann@hzdr.de (A.F.); 2School of Science, Faculty of Chemistry and Food Chemistry, Technical University Dresden, 01062 Dresden, Germany; 3Institute of Biophysics and Radiation Biology, Semmelweis University, 1094 Budapest, Hungary; 4Faculty of Informatik and Wirtschaftsinformatik, Provadis School of International Management and Technology AG, 65926 Frankfurt, Germany; 5National Center for Tumor Diseases (UCC/NCT), Partner Site Dresden, 01307 Dresden, Germany; 6German Cancer Consortium (DKTK), Partner Site Dresden and German Cancer Research Center (DKFZ), 69120 Heidelberg, Germany

**Keywords:** ^177^Lu-labeled antibody, prostate stem cell antigen, CHX-A″-DTPA, prostate cancer

## Abstract

Prostate specific membrane antigen (PSMA) is an excellent target for imaging and treatment of prostate carcinoma (PCa). Unfortunately, not all PCa cells express PSMA. Therefore, alternative theranostic targets are required. The membrane protein prostate stem cell antigen (PSCA) is highly overexpressed in most primary prostate carcinoma (PCa) cells and in metastatic and hormone refractory tumor cells. Moreover, PSCA expression positively correlates with tumor progression. Therefore, it represents a potential alternative theranostic target suitable for imaging and/or radioimmunotherapy. In order to support this working hypothesis, we conjugated our previously described anti-PSCA monoclonal antibody (mAb) 7F5 with the bifunctional chelator CHX-A″-DTPA and subsequently radiolabeled it with the theranostic radionuclide ^177^Lu. The resulting radiolabeled mAb ([^177^Lu]Lu-CHX-A″-DTPA-7F5) was characterized both in vitro and in vivo. It showed a high radiochemical purity (>95%) and stability. The labelling did not affect its binding capability. Biodistribution studies showed a high specific tumor uptake compared to most non-targeted tissues in mice bearing PSCA-positive tumors. Accordingly, SPECT/CT images revealed a high tumor-to-background ratios from 16 h to 7 days after administration of [^177^Lu]Lu-CHX-A″-DTPA-7F5. Consequently, [^177^Lu]Lu-CHX-A″-DTPA-7F5 represents a promising candidate for imaging and in the future also for radioimmunotherapy.

## 1. Introduction

Despite continuous advances in diagnosis and treatment, prostate cancer (PCa) remains a leading cause of mortality for men worldwide [1,2]. Current treatment options, particularly for patients who developed advanced disease after primary surgery or radiation therapy or who have already developed metastases at the time of diagnosis, are usually no more curative and can only delay the progression of the disease. Although hormone therapy can delay PCa, over time, patients frequently develop castration-resistant prostate cancer (CRPC) with metastases in lymph nodes and bones. The median survival for patients with CRPC is 16 to 18 months [3]. Consequently, there is still a tremendous need for more efficient treatment options, particularly for high-risk patients and those with already confirmed metastasized PCa. Moreover, improved methods for detection of PCa metastases are required, providing the basis for accurate diagnosis and recommendation of personalized treatment strategies [4,5]. Even until today, diagnosis of PCa depends on histological analyses. Non-invasive imaging modalities such as magnetic resonance imaging, ultrasound, and nuclear medicine procedures can support the diagnosis. Highly sensitive PET-diagnostics (see below) or scintigraphic diagnostics with ^99m^Tc-labeled methylene diphosphonate (MDP) bone scintigraphy can support the diagnosis, especially for detection of tumor infiltration and metastases. However, each of these techniques has its limitations [6,7].

Given the intrinsic radiosensitivity of most PCa, several targeted radionuclide therapies have been developed [4]. Promising clinical results have been obtained for the lutetium-177 radiolabeled urea derivative lutetium-177–PSMA-617 (^177^Lu-PSMA-617, also known as lutetium-177 vipivotide tetraxetan, Pluvicto^®^) [8,9,10,11]. As shown by the phase III VISION trial (NCT03511664), PSMA-617 labeled with the theranostic radionuclide ^177^Lu leads to a significant reduction in bone- and lymph node metastases and increased the survival rate when applied in combination with standard-of-care therapies, which finally led to its approval thus far for patients with advanced prostate cancer [8,9,10,11]. Moreover, when the urea-derivative is combined with a diagnostic radionuclide, such as ^68^Ga or ^18^F, it is possible to non-invasively characterize PCa using positron emission tomography (PET) [12,13,14,15,16]. Furthermore, fusion of a peptide-tag to the urea-based ligand of PSMA converts the radiotracer into a radio-/immunotheranostic targeting molecule (TM) which can be used for both PET imaging of PCa patients and retargeting of T cells genetically modified with a universal chimeric antigen receptor (UniCAR) for immunotherapy [17,18,19,20].

On the one hand, the performance of these radiolabeled PSMA ligands demonstrate the high potential of a targeted radionuclide therapy for advanced PCa and encourage the theranostic concept that may lead to a substantial improvement in the clinical management of PCa [21]. On the other hand, PSMA is not expressed in all tumor cells and, even worse, it is also expressed in some healthy tissues including for example in fibers of the sympathetic trunk, which may be labelled and misinterpreted as lymph node metastases [22]. This can lead to false positive diagnosis and may influence the therapeutic regimen [23,24,25]. Furthermore, the PSMA levels in PCa do not correlate due to tumor heterogeneity [26]. Since PSMA radioligands accumulate also in non-prostate tissues and lesions, it is important to carefully interpret PSMA images [27]. Even more critical, therapeutic interventions based on PSMA ligands can lead to toxic side effects, particularly the destruction of the salivary glands [28]. Therefore, an additional or alternative prostate specific target would be useful for diagnostic imaging and therapy of PCa.

Prostate stem cell antigen (PSCA) is such a potential tumor target as it is overexpressed in PCa. Interestingly, PSCA is also overexpressed in a variety of other tumor entities including gallbladder-, urinary bladder-, breast-, and pancreatic cancer, as well as renal cell carcinomas and gliomas [25,26,27,28,29,30,31,32,33]. Moreover, expression of PSCA in PCa increases with the Gleason Score, tumor stage, androgen-independent progression, and metastases formation in bone, lymph nodes, or liver [32,34,35]. Like PSMA and all other tumor associated antigens (TAAs), PSCA is not exclusively expressed in tumor tissues. Expression of PSCA was also detected at both the mRNA and protein level in healthy tissues. At the mRNA level, PSCA expression could be detected in prostate, placenta, kidney, and urogenital tissues. At the protein level, expression of PSCA was found in prostate epithelium, epithelial layers of the urinary bladder, neuroendocrine cells of the stomach and colon, collecting ducts of the kidney, trophoblasts of the placenta, and with conflicting reports in pancreas [25,26,27,28,29,30,31,32,33,34,35].

Until now, we and others have established a variety of antibody-based recombinant derivatives against PSCA for immunotherapeutic approaches. For example, we constructed chimeric antigen receptors (CARs) directed against PSCA [20,36,37,38], bispecific anti-CD3-anti-PSCA Abs [39], or monospecific anti-PSCA Ab fragments [40,41,42]. Besides conventional anti-PSCA CARs, we developed modular, and thereby switchable, CAR platforms (UniCARs, RevCARs) that can be used for retargeting of T cells against PSCA positive PCa cells [18,19,20,36,37,38,43,44,45].

In the present study, we conjugated the previously described full size murine anti-PSCA-specific IgG1 mAb 7F5 [40] with the bifunctional chelator p-SCN-CHX-A″-DTPA and radiolabeled the anti-PSCA mAb with the theranostic radionuclide ^177^Lu. After evaluation of the radiolabeled anti-PSCA mAb 7F5 termed [^177^Lu]Lu-CHX-A″-DTPA-7F5, we determined its in vitro properties and biodistribution in mice. Moreover, we performed single photon emission computed tomography (SPECT)/CT. According to the presented data, [^177^Lu]Lu-CHX-A″-DTPA-7F5 represents a promising candidate for imaging and radioimmunotherapy of PCa.

## 2. Results

### 2.1. Purification of the Anti-PSCA mAb 7F5 and Evaluation of Its Specific Binding Capability

After purification of the anti-PSCA mAb 7F5 from hybridoma supernatant by protein A affinity chromatography, dialyzed elution fractions were analyzed by SDS-PAGE (Figure 1(AI)) and immunoblotting using an anti-mouse IgG Ab conjugated with alkaline phosphatase ((Figure 1(AII)). SDS-PAGE was performed under non-reducing and reducing conditions (Figure 1A). As expected, both conditions resulted in a profile typical for an antibody. (i) Under non-reducing conditions, we detected a single protein with a mobility according to a molecular weight of 150 kDa (Figure 1(AI), left panel, 150 kDa) supporting the purity of the isolated antibody fraction. (ii) Under reducing conditions, the disulfide bridges between the heavy and light chains of the ab were split, resulting in two protein bands according to mobilities of 25 and 50 kDa typical for heavy and light chains of an antibody (Figure 1(AI), right panel, and AII, 25 and 50 kDa). Moreover, both protein bands could be detected by an anti-Ig-alkaline phosphatase conjugate (Figure 1(AII), 25 and 50 kDa). The isolated antibody fraction does not contain protein aggregates which would be detectable at the border between the stacking and separation gel.

The binding capability and specificity of purified anti-PSCA mAb 7F5 to PSCA was analyzed by flow cytometry using PSCA-positive PC3 (Figure 1B, PC3-P) or PSCA-negative PC3 cells (Figure 1B, PC3). As shown in Figure 1B, the anti-PSCA mAb 7F5 is able to bind to the PSCA-positive PC3-P cells, but not to the PSCA-negative PC3 cells. By applying increasing concentrations of the anti-PSCA mAb 7F5 to PC3-P cells, an apparent K_D_ value of 10.5 ± 4.0 nM (Figure 1C) was calculated.

### 2.2. Effect of the Anti-PSCA mAb 7F5 on the Viability of PSCA-Positive Cells

Next, we analyzed whether the anti-PSCA mAb 7F5 has a direct effect on the viability of PSCA-positive cells. For this purpose, PC3-P and PC3 cells were incubated for 4, 24, and 48 h with the anti-PSCA mAb 7F5 at two concentrations (6.8 nM and 1000 nM). As positive cytotoxic control served a treatment with Triton X-100 (2.5%). Viability was estimated using the MTS assay as described under Section 4. As shown in Figure 2, the anti-PSCA mAb 7F5 showed no influence on the viability of the cells.

### 2.3. Conjugation of CHX-A″-DTPA to the Anti-PSCA mAb 7F5

For the purpose of ^177^Lu-labeling, p-SCN-Bn-CHX-A″-DTPA was used to couple the chelating moiety CHX-A″-DTPA to primary amino groups of the anti-PSCA mAb 7F5 via its isothiocyanate functional group. MALDI-ToF-MS analyses and HPLC-SEC profiles indicated a successful conjugation of the chelator to the anti-PSCA mAb 7F5. MALDI-ToF-MS analysis of untagged and CHX-A″-DTPA-tagged mAb 7F5 showed comparable MS profiles (Figure 3A). The relative mass difference was used for calculation of the dominant conjugation degree. According to this measurement, the anti-PSCA mAb 7F5 was tagged with an average of four CHX-A″-DTPA moieties. HPLC-SEC profiles of untagged anti-PSCA mAb 7F5 and CHX-A″-DTPA-tagged 7F5 are presented in Figure 3B. Since the anti-PSCA mAb 7F5 was conjugated randomly to the ε-amino groups of lysine residues, the conjugation reaction resulted in a mixture of molecules with a different number of tagged chelator moieties.

### 2.4. Radiolabeling and Stability of the Radiolabeled Ab Conjugate

After ^177^Lu-labeling of CHX-A″-DTPA-7F5, radio-iTLC analysis indicated that [^177^Lu]Lu-CHX-A″-DTPA (derivatives) resulting from excess p-SCN-Bn-CHX-DTPA from the coupling reaction was readily removed by the following centrifugation steps. Finally, a radio-HPLC-SEC was used to detect radiolabeled mAb-aggregates or fragments and to determine the radiochemical purity (RCP). The HPLC profiles of [^177^Lu]Lu-CHX-A″-DTPA-7F5 showed a main peak at a retention time of around 11 min before and after three spin filtration steps while [^177^Lu]Lu-CHX-A″-DTPA eluted around 16 min (Figure 4A). The achieved RCP was usually 95% or more; the specific activity of [^177^Lu]Lu-CHX-A″-DTPA-7F5 was 350 GBq/mg.

According to Lindmo et al. [46], the immunoreactive fraction (IRF) of [^177^Lu]Lu-CHX-A″-DTPA-7F5, applied in a low concentration, is the part of intact specific binding to an infinite amount of PSCA. Using PC3-P cells, we determined an averaged IRF of 62% (Figure 4B). As expected, an IRF value could not be determined for PSCA negative PC3 cells (Appendix A).

To determine its stability, [^177^Lu]Lu-CHX-A″-DTPA-7F5 was separated via SDS-PAGE two days after radiolabeling (Figure 5A, panel a,b) under reducing (left lanes) and non-reducing conditions (right lanes). According to silver staining (Figure 5A, panel a) and autoradiography (Figure 5A, panel b), the [^177^Lu]Lu-CHX-A″-DTPA-7F5 is stable over 2 days. The stability was confirmed by incubation of [^177^Lu]Lu-CHX-A″-DTPA-7F5 in PBS, cell medium and human serum for 0, 1, and 2 days (Figure 5B). After each time point, the intact radiolabeled Ab was absorbed on a protein A affinity column, whereas radioactive constituents, which were released from the Ab (e.g., by transchelation) or smaller Ab fragments, did not bind to the column and are found in the flow through. The stability represents the portion of protein A bound radioactivity.

### 2.5. In Vitro Binding Characteristics

The time-dependent uptake of [^177^Lu]Lu-CHX-A″-DTPA-7F5 was determined for PC3-P cells. The internalization in relation to the sum of binding and internalization was less than 10% within the first 4 h (8.6% at 37 °C, 5.9% at 4 °C). The uptake increased 32.4% and 44.8% after 24 h and 48 h, respectively (Figure 6A). Consequently, saturation binding studies could be performed at 37 °C on intact cells, since internalization within 2 h was low at this temperature (see Figure 6A). The [^177^Lu]Lu-CHX-A″-DTPA-7F5 binding on PC3-P cells resulted in an equilibrium dissociation constant K_D_ of 32.6 ± 12.1 nM and a binding capacity B_max_ of 13.8 ± 2.6 pmol/mg protein at 37 °C based on the data shown in Figure 6B. In control assays using the PSCA-negative control cell line PC3, we observed no specific binding and no specific uptake of [^177^Lu]Lu-CHX-A″-DTPA-7F5 (Appendix A).

### 2.6. In Vivo Targeting

#### 2.6.1. Biodistribution Studies

The results of the biodistribution, 1 h and 48 h after intravenous injections of [^177^Lu]Lu-CHX-A″-DTPA-7F5 in athymic mice bearing PSCA expressing tumors, are shown in Figure 7 (Appendix A).

One hour after intravenous administration, the SUV value (standard uptake value normalized to the body weight) of blood reached 6.6 (30.3% ID/g see Appendix A). Moreover, a relative high uptake of [^177^Lu]Lu-CHX-A″-DTPA-7F5 was observed in tissues and organs that are well vascularized, as well as in organs involved in biotransformation and excretion, such as the heart, lungs, liver, kidneys, and spleen. After 48 h, the [^177^Lu]Lu-CHX-A″-DTPA-7F5 accumulation significantly decreased in the blood, heart, brain, lungs, adrenals, kidney, and brown adipose tissue, whereas it increased in the liver, as well as the skin and hair, but particularly in the tumor (Figure 7A). Tumor uptake reached an SUV value of 3.1 (14.5% ID/g see Appendix A) after 48 h and was significantly higher compared to normal tissues, except for liver and spleen. The similar uptake of [^177^Lu]Lu-CHX-A″-DTPA-7F5 in liver and intestine, and kidneys and urine after 1 h (18 ± 2 vs. 19 ± 1% ID) suggests an initial hepatobiliary and renal elimination route at equal parts (Figure 7B). Two days later, the excretion occurred more via the renal pathway (21 ± 4 vs. 27 ± 1% ID). Furthermore, both the tumor/muscle and the tumor/blood ratio were significantly increased from 1 h to 48 h (Figure 7C).

#### 2.6.2. Small Animal Imaging

Both PSCA-positive PC3-P and PSCA-negative PC3 tumors were established in the same mice. The tumor xenograft-bearing mice were examined using SPECT/CT from 1 h to 6 days after a single intravenous injection of [^177^Lu]Lu-CHX-A″-DTPA-7F5 (Figure 8, 32 MBq/170 pmol). SPECT/CT images for one out of two representative mice are shown in Figure 8A indicating a specific uptake of [^177^Lu]Lu-CHX-A″-DTPA-7F5 in the PSCA-positive PC3-P tumor, while no uptake was observed in the PSCA-negative PC3 tumor. However, SUV in heart (blood), liver, and spleen decreased over time, uptake of [^177^Lu]Lu-CHX-A″-DTPA-7F5 significantly increased in PC3-P (Figure 8C). At 144 h p.i. tumor SUV was 0.5 ± 0.1 and 5.7 ± 0.6 for PC3 and PC3-P tumors, respectively, which led to a PC3-P/PC3 ratio of 11.3 ± 2.3 and a PC3-P/blood ratio of 6.1 ± 0.9 at 144 h p.i. (Figure 8D). The detectable limited label in the spine may be due to metastasizing tumor cells (Figure 8B).

## 3. Discussion

PSCA represents a potential target for imaging and radio-/immunotherapy of advanced PCa, but also other solid tumors including bladder-, breast-, and pancreatic cancer, and brain tumors (e.g., [25,26,27,28,29,30,31,32,33]). Previously, we described the anti-PSCA mAb 7F5 [40] and its potential application for immunotherapy using either a bispecific antibody or a CAR T cell approach (e.g., [20,39]). The major aim of this study was to radiolabel the anti-PSCA mAb 7F5 and evaluate its potential use for imaging, radio-/immunotherapy and combinations thereof.

For the purified anti-PSCA mAb 7F5, we confirmed the specific binding and estimated a high affinity towards PSCA with a K_D_ value in the low nM range as shown by flow cytometry comparing PSCA-positive (PC3-P) and PSCA-negative PC3 cells (PC3). In contrast to another previously reported anti-PSCA mAb (1G8) [47,48,49], the anti-PSCA mAb 7F5 did not influence the proliferation and metabolic activity of PCa cells with and without PSCA surface expression.

For radiolabeling of the anti-PSCA mAb 7F5, we selected the radionuclide ^177^Lu as its relatively long half-life (t_1/2_ = 6.73 d) is in accordance with the pharmacokinetic of mAbs and offers advantages over other β^−^-emitters, e.g., ^90^Y, with regard to patient logistics [50,51]. Furthermore, ^177^Lu emits γ-photons detectable in biodistribution and imaging studies using SPECT. In addition, ^177^Lu is a medium-energy β^−^-emitter which is suitable for the treatment of small tumor lesions (1.2–3.0 mm), e.g., close to the bone marrow or liver, a common situation in metastatic PCa [4,52,53].

For chelating of the radionuclide ^177^Lu we conjugated the acyclic chelator CHX-A″-DTPA via p-SCN-Bn-CHX-A″-DTPA to the anti-PSCA mAb 7F5 [54]. The chelator was randomly coupled to primary amines of the anti-PSCA mAb 7F5 via the functional isothiocyanate group. According to MALDI-ToF-MS analysis, the anti-PSCA mAb 7F5 was conjugated with an average of four CHX-A″-DTPA moieties. In other studies, the macrocycle DOTA is used as chelating agent for labeling of mAbs with ^177^Lu. The acyclic chelating moiety CHX-A″-DTPA, however, may be superior compared to DOTA as the chelation can be achieved under mild reaction conditions (1 h, 23 °C or 37 °C, in aqueous buffer system at pH 5), which is an obvious advantage for the labeling of temperature sensitive proteins such as Abs [55,56,57,58]. Indeed, the labeling of the anti-PSCA mAb 7F5 with ^177^Lu resulted in a high average specific activity in the range of 350 MBq/mg. Since free ^177^Lu accumulates in the skeleton [59], we determined the radiochemical stability of the ^177^Lu-labeled radioimmunoconjugate [^177^Lu]Lu-CHX-A″-DTPA-7F5. The data measured in PBS, cell media and human serum suggest a high stability of [^177^Lu]Lu-CHX-A″-DTPA-7F5. These data are in line with other ^177^Lu-labeled Abs conjugated with the CHX-A″-DTPA moiety [58,60,61,62,63]. Moreover, we observed no enrichment in the skeleton during SPECT imaging. After ^177^Lu-labeling, the IRF of [^177^Lu]Lu-CHX-A″-DTPA-7F5 was slightly decreased. This could be caused by the random coupling of the chelating moieties in case coupling happened in the antigen-binding region. Still the remaining functional portion of [^177^Lu]Lu-CHX-A″-DTPA-7F5 is in the range of other CHX-A″-DTPA-mAb-conjugates labeled with ^177^Lu [63,64,65,66]. According to our saturation binding assays, the modification and radiolabeling of the anti-PSCA mAb 7F5 [^177^Lu]Lu-CHX-A″-DTPA-7F5 does not destroy its high binding affinity with a K_D_ of about 30 nM. Most important, there was only a specific binding to PSCA-overexpressing PC3 cells but no binding to PSCA-negative PC3 cells as measured by flow cytometry and finally supported by the SPECT imaging mouse data. Although there was a slight loss of affinity compared to the unlabeled anti-PSCA mAb 7F5, the remaining binding affinity is just in the range that is considered to be favorable for tumor-targeting IgG antibodies since very high binding affinities of radioimmunoconjugates (K_D_ values < 1 nM) might be associated with low penetration efficiency and limited diffusion rate into tumors due to the so-called binding site barrier effect, which can reduce the therapeutic efficiency [67,68].

It is well accepted that an internalization of antibodies does not occur at 4 °C [69,70]. In a previous publication, it was, however, shown that the analyzed anti-PSCA-Ab can be internalized via the caveolae pathway and accumulate in endolysosomal compartments under physiological conditions [71]. In contrast, for the radiolabeled mAb 7F5, we observed a specific and rather stable binding to the surface of the PSCA-positive PC3 cells PC3-P. No specific binding could be detected for PSCA-negative PC3 cells. Obviously, a stable surface binding is favorable for immunotherapeutic approaches and also, most likely, for an alternative treatment approach, known as pretargeting [72,73,74].

In summary, here, we described the successful radiolabeling of the monoclonal anti-PSCA antibody 7F5 with ^177^Lu and its use for SPECT imaging. The highly specific and stable binding to PSCA positive tumor cells makes the anti-PSCA antibody 7F5 of high interest both for imaging and radioimmunotherapy e.g., of PSMA-negative PCas.

## 4. Materials and Methods

### 4.1. Cell Culture

If not noted otherwise, all cells were cultured in RPMI 1640 medium supplemented with 10% fetal calf serum (FCS), 100 U/mL penicillin and 100 μg/mL streptomycin, 2 mM N-acetyl-L-alanyl-L-glutamine, 1% non-essential amino acids (NEA), and 1 mM sodium pyruvate (all Biochrom GmbH, Berlin, Germany). All cells were maintained at 37 °C in a humidified atmosphere with 5% CO_2_. Adherent cells were detached with 2 mM EDTA in Dulbecco’s phosphate-buffered saline (PBS) w/o Ca^2+^ and Mg^2+^ (Biochrom GmbH).

#### 4.1.1. PCa Cell Lines

Binding of the anti-PSCA mAb was investigated using the PCa cell lines. As PCa cell lines usually downregulate the expression of PSCA in cell culture, we used PC3 cells genetically modified to overexpress human PSCA as described previously (e.g., [41]). The cell lines overexpressing PSCA were termed as PC3-P. The genetic manipulation leads to a reliable expression at the moderate level of around 25 × 10^3^ PSCA molecules on the surface per cell.

#### 4.1.2. Hybridoma Cell Line

The generation of hybridoma cells producing the anti-PSCA mAb 7F5, which is specific for human cell surface-localized PSCA, has been described previously [40]. The cells were expanded to terminal cultures in the same medium as the PCa cells with additional 0.5% β-mercaptoethanol. Later, continuous Ab production was accomplished with the same cell medium containing only 0.5% FCS.

### 4.2. Purification of the Anti-PSCA mAb 7F5 from Hybridoma Cell Culture Supernatant

Every 2 to 3 days, 80 to 90% of the hybridoma cell culture supernatant was harvested and replaced with fresh medium. The supernatant was centrifuged at 300× *g* for 5 min. The resulting supernatant was cleared from cell debris by an additional centrifugation step (10,000× *g* for 10 min) and filtrated (0.22 µm pore size). All steps were performed at 4 °C. The salt concentration in the final supernatant was raised by addition of solid NaCl (final concentration 1 M) to improve the binding of the IgG mAb to the protein A Sepharose matrix. The antibody solution was buffered by addition of 3 M Tris-HCl (47 mM Tris/HCl, pH 8.8) and purified by affinity chromatography using a protein A Sepharose column (Amersham Biosciences AB, Uppsala, Sweden). Before elution, the column was washed with PBS (w/o Mg^2+^ and Ca^2+^) containing Tris/HCl (47 mM Tris/HCl, 1 M NaCl, pH 8.8). Elution was performed using glycine/HCl (pH 3.0, 0.15 M NaCl). Besides monitoring pressure and pH during the elution, the ÄKTAexplorer 10 C 205 chromatography system (Cytiva Europe GmbH, Freiburg, Germany) was used to detect the Ab containing fractions via UV/Vis detector. Fractions containing the anti-PSCA mAb 7F5 were pooled and dialyzed against 0.05 M HEPES (pH 6.0, 0.3 M NaCl) for storage or 0.05 M NaHCO_3_ (pH 6.4, 0.3 M NaCl) for subsequent conjugation with p-SCN-CHX-A″-DTPA. Dialysis was performed with a porous membrane tubing (MWCO 100 kDa) 15 h at 4 °C under continuous stirring (in 1 L buffer, once buffer change). The protein content was measured by UV absorbance at 280 nm (Specord 210, Analytik Jena AG, Jena, Germany) using a molar attenuation coefficient (ε) of 233,605 L∙mol^−1^∙cm^−1^.

### 4.3. Ab Purity and Initial Characterization

The Ab purity was verified by SDS-PAGE (sodium dodecyl sulfate polyacrylamide gel electrophoresis) (12% separation gel, 5% stacking gel) and subsequent staining of the gel with Coomassie Brilliant Blue G-250 (Thermo Fisher Scientific GmbH, Schwerte, Germany). Western blotting (wet tank transfer system) was performed on a nitrocellulose membrane (Hybond^TM^ ECL^TM^, VWR International GmbH, Darmstadt, Germany). After blocking, Western blots were analyzed using an alkaline phosphatase conjugated rabbit-Ab against murine immunoglobulin G (IgG) (Dianova, Berlin, Germany).

Size-exclusion (SEC)-HPLC was performed to detect potential Ab aggregates. For this purpose, an Agilent Bio SEC-3 column (3 µm, 150 Å, 7.8 ID × 150 mm; Agilent, Santa Clara, CA, USA) was run at 1.0 mL/min flow and with 0.1 M sodium phosphate buffer (pH 7.0) as mobile phase. HPLC was performed using a Chromaster HPLC-System 600 (VWR Hitachi, Darmstadt, Germany).

### 4.4. Binding Analysis by Flow Cytometry

The specific binding capability of the anti-PSCA mAb 7F5 was evaluated by flow cytometry analysis. Briefly, 5 × 10^5^ PC3-P or PC3 cells were suspended in 90 μL PBS containing 2% FCS and incubated on ice with 10 μL of the anti-PSCA mAb 7F5 (150 nM final concentration). As control for non-PSCA but Ig related antibody binding, cells were incubated with the anti-EGFR specific IgG1 Ab (Cetuximab, Merck KGaA, Darmstadt, Germany). After 1 h, the cells were washed and incubated with phycoerythrin-conjugated goat F(ab’)_2_ fragment anti-mouse IgG (fcg) (1:400), Beckmann Coulter, Krefeld, Germany) at 4 °C for 30 min. Cells were washed again, pelleted, and resuspended in 10 µL PBS with 2% FCS and 90 µL propidium iodide (1.5 µM). Binding data were acquired on a MACSQuant^®^ Analyzer 10 and analyzed by using MACSQuantify^®^ software 2.1 (Miltenyi Biotec GmbH, Bergisch-Gladbach, Germany). The apparent binding affinity of the anti-PSCA mAb 7F5 was determined by incubation of tumor cells with ten different concentrations of the anti-PSCA mAb 7F5 ranging between 0.5 and 145 nM.

### 4.5. Cell Proliferation Assay

In order to estimate the influence of the anti-PSCA mAb 7F5 on the viability of PC3-P and PC3 cells, an MTS (3-(4,5-dimethylthiazol-2-yl)-5-(3-carboxy methoxyphenyl)-2-(4-sulfophenyl)-2H-tetrazolium) assay was performed using the CellTiter 96^®^ Aqueous One Solution Cell Proliferation Assay (Promega GmbH, Mannheim, Deutschland). For this purpose, 2500 cells/well were seeded and cultured in 100 µL of media in 96-well-plates. After 24 h, the cells were treated with two different concentrations of the anti-PSCA mAb 7F5 (7.8 nM or 1 µM). As cytotoxic control, we used a treatment with Triton X-100 (2.5%). After 4, 24, and 48 h, the viability was determined according to the manufacturer’s instructions. Absorption was measured at 490 nm using a spectral multimode well-plate reader (Varioskan^®^Flash, Thermo Fisher Scientific GmbH).

### 4.6. Ab Conjugation with p-SCN-CHX-A″-DTPA

The anti-PSCA mAb 7F5 was modified with p-SCN-Bn-CHX-A″-DTPA (Macrocyclics, Plano, TX, USA) by coupling the functional isothiocyanate group to primary amines in the protein structure. First, the mAb was concentrated to about 1.0 mg/mL in a MWCO 30 kDa MacrosepTM centrifugal concentrator (PALL, Port Washington, NY, USA). Thereby, the buffer was exchanged with 0.05 M NaHCO_3_ (pH 6.4, containing 0.3 M NaCl) using the same centrifugal technique. For modification of 5 mg mAb, 4.4 mg p-SCN-Bn-CHX-A″-DTPA (185-fold molar excess) was dissolved in three times the volume 0.5 M HEPES (pH 7.2, containing 0.3 M NaCl) and added dropwise to the Ab solution. The mixture was incubated for 24 h at 23 °C with occasional slewing. The resulting conjugate was separated from excess of p-SCN-Bn-CHX-A″-DTPA and other reactants by repeated washing (6×) using, in total, 500 mL of 0.05 M HEPES (pH 6.0, containing 0.3 M NaCl) in a MWCO 30 kDa JumbosepTM centrifugal concentrator (Pall, Port Washington, NY, USA). SEC-HPLC of the conjugate was performed as described above for the unconjugated Ab.

### 4.7. Mass Spectroscopy of the Conjugate

Matrix-assisted laser desorption/ionization time of flight (MALDI-TOF) mass spectrometry was performed to determine the average number of chelator moieties per Ab molecule. The matrix was a 2,5-dihydroxyacetophenone (DHAP)-based solution consisting of 7.6 mg 2,5-DHAP for MALDI-MS (Merck) in 375 μL absolute ethanol with 125 μL of an 18 mg/mL aqueous solution of diammonium hydrogen citrate. The sample was mixed with 2% trifluoroacetic acid and matrix solution at equal volumes and pipetted up and down until crystallization started. Then, 0.5 µL crystalized sample was added on the steel ground target and allowed to dry. Measurements were carried out with an Autoflex II TOF/TOF mass spectrometer (Bruker, Billerica, MA, USA) and analyses of the spectra with flexAnalysis software 3.4.

### 4.8. ^177^Lu-Labeling of [^177^Lu]Lu-CHX-A″-DTPA-7F5

Non-carrier added [^177^Lu]LuCl_3_ (EndolucinBeta^®^, in 0.04 M HCl) was purchased from ITM (Isotopen Technologien München AG, Munich, Germany). Radiolabeling of CHX-A″-DTPA-7F5 with ^177^Lu was achieved by mixing about 50 MBq of the radionuclide solution with the same volume of 0.05 M HEPES (pH 6.0, containing 0.3 M NaCl) to obtain a pH between 5 and 6. Then, the solution was added to 1 nmol of CHX-A″-DTPA-7F5 conjugate and allowed to react for 60 min at 37 °C. An amount of 1 nmol CHX-A″-DTPA was added in order to complex free ^177^Lu for 10 min at 23 °C. The reaction mixture was then separated via spin filtration with a 30 kDa MWCO Amicon Ultra-0.5 mL centrifugal filter (Merck).

Residual free ^177^Lu in the [^177^Lu]Lu-CHX-A″-DTPA-7F5 preparation was determined by using instant thin layer chromatography (iTLC) with a silica gel-impregnated glass fiber support and a mobile phase of 10 g/L DTPA in H_2_O (23.2 mM, pH 5.5) and autoradiographic analysis (BAS-3000, Fuji, Japan). Radiochemical purity was determined using SEC-HPLC with a TSKgel SuperSW mAb HR column (4 µm, 250 Å, 7.8 ID × 300 mm; Tosoh, Tokyo, Japan) and 0.2 M sodium phosphate buffer (pH 6.7, containing 0.1 M Na_2_SO_4_) as mobile phase at 0.8 mL/min flow. HPLC was performed using a Knauer HPLC-System (Knauer Wissenschaftliche Geräte GmbH, Berlin, Germany) with a Ramona detector (Elysia-raytest GmbH, Angleur, Belgium) for radioactivity flow monitoring.

### 4.9. Stability Studies

Approximately 100 kBq of [^177^Lu]Lu-CHX-A″-DTPA-7F5 or [^177^Lu]Lu-p-SCN-CHX-A″-DTPA was mixed 1:10 with PBS, cell medium, or human serum and maintained at 37 °C. After 1, 24, and 48 h, samples were mixed 1:10 with protein A binding buffer and thereafter absorbed on a pre-equilibrated protein A affinity column. Released radioactive constituents were compared with the bound activity that was then eluted from the column. Ab integrity was further determined using SDS-PAGE (12% separation gel, 5% stacking gel) with subsequent autoradiography (Typhoon FLA 9500; GE Healthcare, Little Chalfont, GBR) and silver staining (Pierce Silver Stain Kit; Thermo Fisher Scientific GmbH) of the gel.

### 4.10. Radioligand Binding Studies

#### 4.10.1. Immunoreactivity

The immunoreactive fraction (IRF) of [^177^Lu]Lu-CHX-A″-DTPA-7F5 was assessed using the method of Lindmo et al. [46], where binding of a low radiolabeled mAb concentration is determined at infinite excess of antigen. For the non-specific binding, preincubation with 500 nM cold anti-PSCA mAb 7F5 in 0.05 M HEPES (pH 6.4, containing 0.3 M NaCl) was performed for 1 h at 37 °C. For total binding, the cells were incubated with corresponding vehicle (0.05 M HEPES, pH 6.4, containing 0.3 M NaCl). Subsequently, the samples were incubated with 1 nM [^177^Lu]Lu-CHX-A″-DTPA-7F5 for 2 h at 37 °C, filtered through glass fiber filters (GF/C, Whatman, VWR), and finally treated with 0.3% polyethylenimine for 90 min using a cell harvester (Brandel, Gaithersburg, MD, USA). After washing with ice-cold PBS, the radioactivity accompanied with the cells harvested on individual filter punches was measured using an automatic gamma counter (1480 WIZARD 3′′, PerkinElmer, Waltham, MA, USA) together with standards containing the total activity. Data were plotted as reciprocal of the cell concentration (X axis) against the quotient of totally applied and bound radiolabeled mAb (Y axis) and fitted with linear regression using the nonlinear regression analysis software GraphPad Prism 8 (GraphPad Software Inc., La Jolla, CA, USA). The Y intercept value gave the reciprocal of the IRF (%).

#### 4.10.2. Internalization

Cells were incubated with [^177^Lu]Lu-CHX-A″-DTPA-7F5 (final concentration 10 nM) for 0 h (5 min), 1, 4, 24, and 48 h at 37 °C, respectively. Adjacent samples received 0.5 to 1 µM of cold anti-PSCA mAb 7F5 1 h before radioligand application to obtain the non-specific binding. After each time interval, the incubation medium was removed, and the cells were washed three times with ice-cold PBS (containing Mg^2+^ and Ca^2+^). Surface-bound activity was stripped with 4 °C-cold acid-wash buffer (0.1 M glycine, 0.15 M NaCl, pH 3.0) for 5 min. Cytosolic activity was determined after treatment with cell lysis buffer (1% SDS in 0.1 M NaOH) for 30 min at 23 °C. Cell surface and cytosolic activities were measured separately in a gamma counter.

#### 4.10.3. Saturation Binding Studies

After treating a part of the cell samples with an excess of cold anti-PSCA mAb 7F5 (0.5 µM) for 1 h to obtain the non-specific binding, all cell samples of a plate were incubated with eight concentrations of [^177^Lu]Lu-CHX-A″-DTPA-7F5 (0.5 to 70 nM) for 2 h at 37 °C. Free Ab was removed by washing three times with ice-cold PBS (containing Mg^2+^ and Ca^2+^). The cells were detached from the culture dish by incubation with cell lysis buffer for 30 min at 23 °C. Activity of the cell suspensions was measured in a gamma counter. Specific binding was calculated and normalized to the protein content of the individual samples measured using bicinchoninic acid assay at 562 nm (Pierce BCA Protein Assay Kit, Thermo Fisher Scientific GmbH) with a spectral multimode well-plate reader (Varioskan^®^Flash, Thermo Fisher Scientific GmbH). Data were fitted with the saturation binding model and the K_D_ value was calculated using the nonlinear regression analysis software GraphPad Prism 8.

### 4.11. Animal Studies

All animal handling was performed according to the guidelines of the German Regulations of Animal Welfare. The protocol was approved by the local Ethical Committee for Animal Experiments (reference number 24-9168.21-4/2004-1). For SPECT imaging, an amount of 4 × 10^6^ PC3 and 3 × 10^6^ PC3-P cells was injected subcutaneously in the left and right flank, respectively, of athymic NMRI nude mice (Rj:NMRI-Foxn1nu/nu; Janvier Labs, Le Genest-Saint-Isle, France). For biodistribution, mice were injected with PC3-P cells on their right flank only. Subcutaneous tumors were allowed to grow for about 21 days to reach a tumor size of about 500 mm^3^. Before radioactivity application the mice weighed 23 ± 1.9 g for biodistribution (n = 8) or 27 ± 1.7 g for SPECT imaging (n = 2).

#### 4.11.1. Biodistribution

Biodistribution of [^177^Lu]Lu-CHX-A″-DTPA-7F5 (molar activity 47 GBq/µmol) was evaluated in male NMRI nude mice bearing a PC3-P tumor on their flank. An amount of 0.28 ± 0.02 MBq (12.2 ± 1.1 MBq/kg; 0.26 nmol/kg body weight) of [^177^Lu]Lu-CHX-A″-DTPA-7F5 was administered into tumor-bearing mice via tail vein injection. Organ distribution of [^177^Lu]Lu-CHX-A″-DTPA-7F5 was measured 1 h and 48 h after injection (2 groups, à 4 mice). The mice were sacrificed under deep anesthesia by heart puncture and cervical dislocation; organs and tissues of interest were excised and weighted. The [^177^Lu]Lu-CHX-A″-DTPA-7F5 uptake was measured in comparison to three reference samples in a gamma counter and high activity values in a cross calibrated dose counter. The decay corrected activity concentration in organs and tissues was expressed as percent injected dose per g tissue (%ID/g) and standardized uptake value (SUV = (activity in the organ/weight of the organ)/(injected activity/animal weight)). Accumulated activity counts (%ID) were measured in all fully extractable organs. Intestine and stomach were measured with content (w.c.), and urine plus feces activity was calculated as the difference between the activity recovery and the injected activity.

#### 4.11.2. Small Animal SPECT/CT-Imaging

[^177^Lu]Lu-CHX-A″-DTPA-7F5 (12.5 ± 1.4 MBq, 16.3 nmol/kg body weight) was injected into a lateral tail vain of female NMRI nude mice bearing both PC3 and PC3-P tumors on their flanks under general desflurane anesthesia (10% desfluran in a 30% oxygen/air mixture). SPECT/CT studies were performed between 1 and 144 h after [^177^Lu]Lu-CHX-A″-DTPA-7F5 injection (nanoSPECT/CT, Mediso Medical Imaging Systems, Budapest, Hungary). Imaging data were presented as maximum intensity projection (MIP). Anatomical reference images were captured before SPECT acquisition using CT-subsystem of the nanoSPECT/CT scanner. Imaging data were reconstructed by Tera-Tomo^TM^ 3D SPECT iterative reconstruction (Mediso) using CT images for attenuation correction. The images were prepared using ROVER (ABX GmbH, Radeberg, Germany) and illustrated as MIP.

### 4.12. Statistical Analysis

For statistical analysis, t-tests were performed using the GraphPad Prism software version 8.0 (GraphPad Software Inc.) whereat * *p* < 0.05; ** *p* < 0.01; *** *p* < 0.001.

## 5. Conclusions

The anti-PSCA mAb 7F5 can be successfully radiolabeled with ^177^Lu resulting in a promising tool for imaging and potentially for therapeutic targeting of PSCA positive tumor entities including, for example, PSMA-negative PCas. Biodistribution and SPECT imaging studies indicate a high specific accumulation in PSCA-positive tumors, but little, if any, enrichment in PSCA-negative tumors, and, in addition, an adequate elimination of [^177^Lu]Lu-CHX-A″-DTPA-7F5 via the excretory organs. Due to low, if any, uptake in non-targeted tissues, the ^177^Lu labeled Ab might be effective in visualizing and treatment, especially of metastases. The stable surface binding of the radiolabeled anti-PSCA antibody 7F5 and its specific accumulation at PSCA-positive tumor sites is favorable for application of the anti-PSCA antibody 7F5 and its radiolabeled derivates for imaging, immunotherapies, endoradionuclide therapies, and combinations thereof.

## Figures and Tables

**Figure 1 ijms-24-09420-f001:**
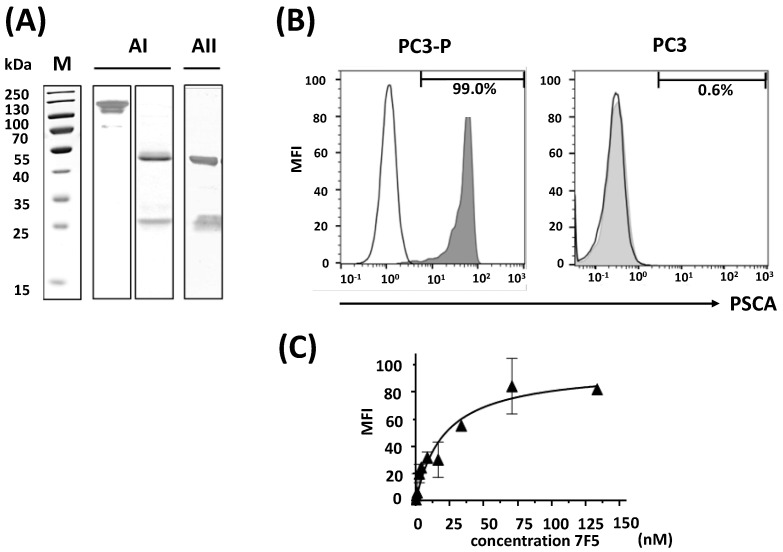
Characterization of purified anti-PSCA mAb 7F5. (**A**) The dialyzed elution fraction of the isolated anti-PSCA mAb 7F5 was separated by SDS-PAGE and either subsequently stained with Coomassie Brilliant Blue G250 (**AI**) or followed by immunoblotting (**AII**). SDS-PAGE was performed under non-reducing ((**AI**), left lane) or reducing conditions ((**AI**), right lane; (**AII**)). After transfer to nitrocellulose membrane (**AII**), the presence of IgG heavy and light chains in the purified anti-PSCA 7F5 mAb fraction was verified using anti-mouse IgG conjugated with alkaline phosphatase (**AII**). M, molecular weight marker. (**B**) To test for specific binding of the anti-PSCA mAb 7F5, FACS analyses were performed to either PSCA-positive PC 3 cells ((**B**), PC3-P, left panel, grey graph) or PSCA-negative PC3 cells ((**B**), PC3, right panel, grey graph). Binding was detected via a phycoerythrin-conjugated goat F(ab’)2 fragment anti-mouse IgG (Fcg). According to previous studies [45], PC3 cells express the epidermal growth factor receptor only at very low level. As negative control, we therefore stained the cells with an anti-EGFR mAb and the phycoerythrin-conjugated goat F(ab’)2 fragment anti-mouse IgG (Fcg) ((**B**), white graphs). Bars represent the percentage of antigen-positive cells. (**C**) Representative specific binding of the anti-PSCA mAb 7F5 to PC3-P cells yielded a K_D_ value of 10.5 ± 4.0 nM (n = 3).

**Figure 2 ijms-24-09420-f002:**
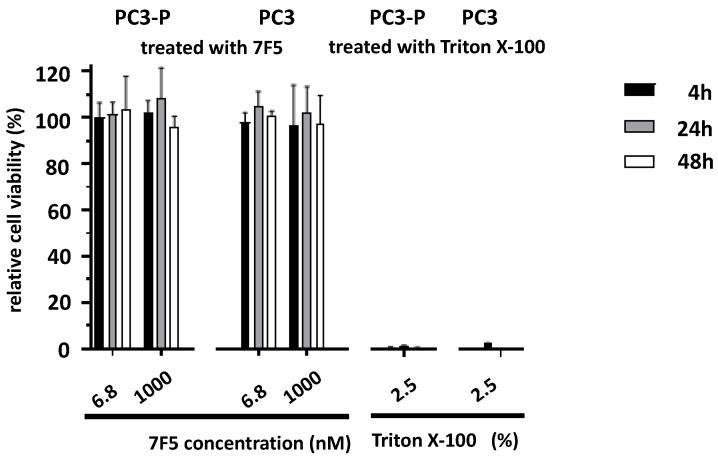
Influence of anti-PSCA mAb 7F5 on the viability of PC3-P or PC3 cells measured by the MTS proliferation test as described under Section 4 after different incubation times (4 h (black), 24 h (gray) and 48 h (white) of treatment); (n = 3). The results were normalized to the untreated medium control and are shown as percentage of living cells; values are averages ± SD from 3 to 4 experiments. As cytotoxic positive control served a treatment with Triton X-100 (2.5%) performed at 4 h (n = 12).

**Figure 3 ijms-24-09420-f003:**
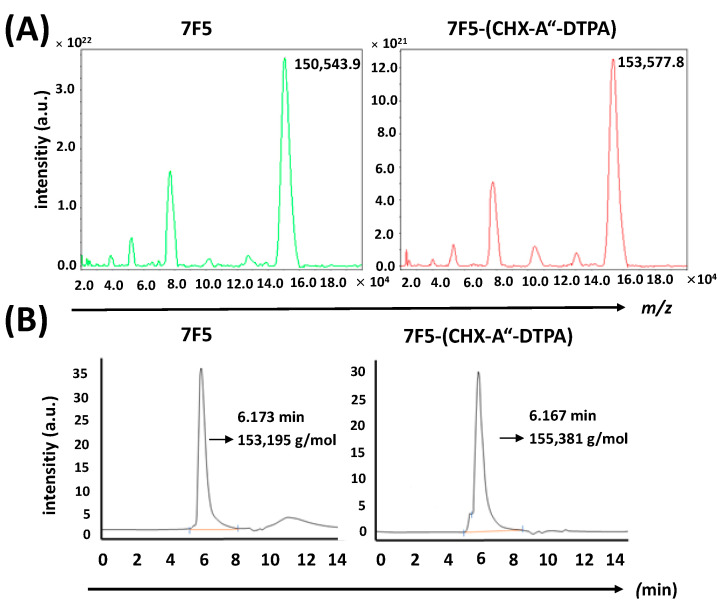
MALDI-ToF and SEC-HPLC analysis of CHX-A″-DTPA-functionalized and non-functionalized anti-PSCA mAb 7F5. (**A**) MALDI-ToF mass spectra of the anti-PSCA mAb 7F5 before [M + H]+ = 150,544 and after conjugation with CHX-A″-DTPA [M + H]+ = 153,578. (**B**) HPLC-SEC elution profiles (UV = 280 nm) of non-conjugated and conjugated anti-PSCA mAb 7F5. Separation was performed using an Agilent Bio SEC-3 column as described under Section 4 (150 Å, 3 µm, 7.8 ID × 150 mm, 0.1 M sodium phosphate buffer pH 7.0, flow 1 mL/min, 23 °C).

**Figure 4 ijms-24-09420-f004:**
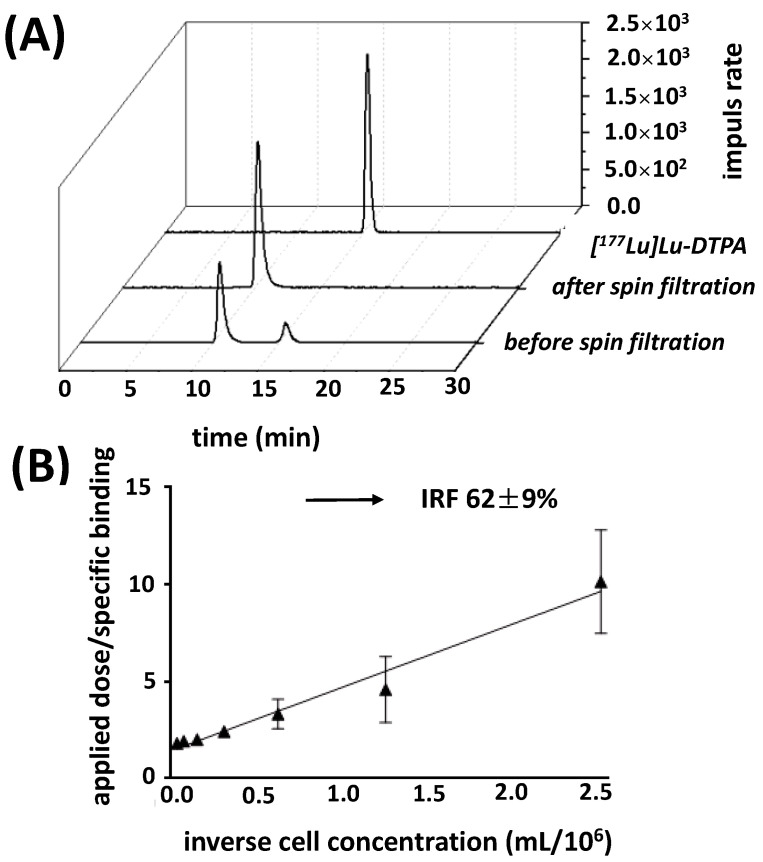
Quality control after ^177^Lu-labeling of 7F5-(CHX-A″-DTPA)_4_. (**A**) For estimation of the radiochemical purity, radio-SEC-HPLC profiles were determined prior and after spin filtration in comparison to [^177^Lu]Lu-CHX-A″-DTPA using a TSKgel SuperSW mAb HR column as described under Section 4. (**B**) IRF of [^177^Lu]Lu-CHX-A″-DTPA-7F5. The radiolabeled Ab was incubated with increasing amounts of PC3-P cells at 37 °C for 2 h. Afterwards, the radiolabeled cells were isolated by filtration through a glass fiber filter. The radioactivity bound to the filter was counted in a gamma counter. Totally applied activity was divided by the specifically bound radioactivity on the cells and then plotted as a function of the reciprocal antigen concentration. The Y intercept represents the reciprocal of the IRF (n = 3, value in % ± SEM).

**Figure 5 ijms-24-09420-f005:**
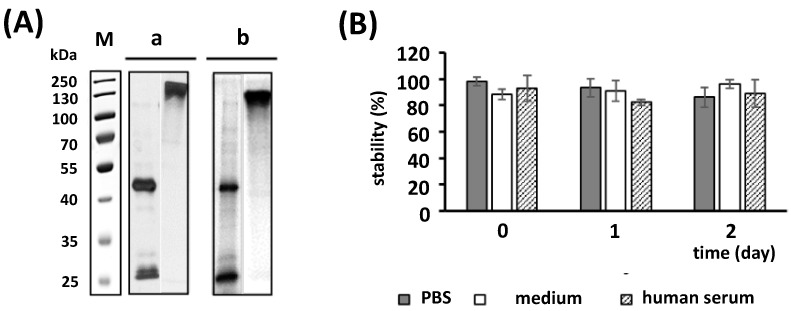
Stability of [^177^Lu]Lu-CHX-A″-DTPA-7F5. (**A**) SDS-PAGE of [^177^Lu]Lu-CHX-A″-DTPA-7F5 two days after radiolabeling. [^177^Lu]Lu-CHX-A″-DTPA-7F5 was separated under reducing ((**a**,**b**), left lanes) and non-reducing ((**a**,**b**), right lanes) conditions. Proteins were detected by either silver staining (**a**) or autoradiography (**b**). M, molecular weight marker. (**B**) Incubation of [^177^Lu]Lu-CHX-A″-DTPA-7F5 with PBS, cell culture medium, or human serum for 0, 1, or 2 days. After each time point, the intact amount of radiolabeled Ab was absorbed on a protein A affinity column and measured as described under Section 4. The stability represents the portion of protein A bound activity (n = 3, stability (%) ± SD).

**Figure 6 ijms-24-09420-f006:**
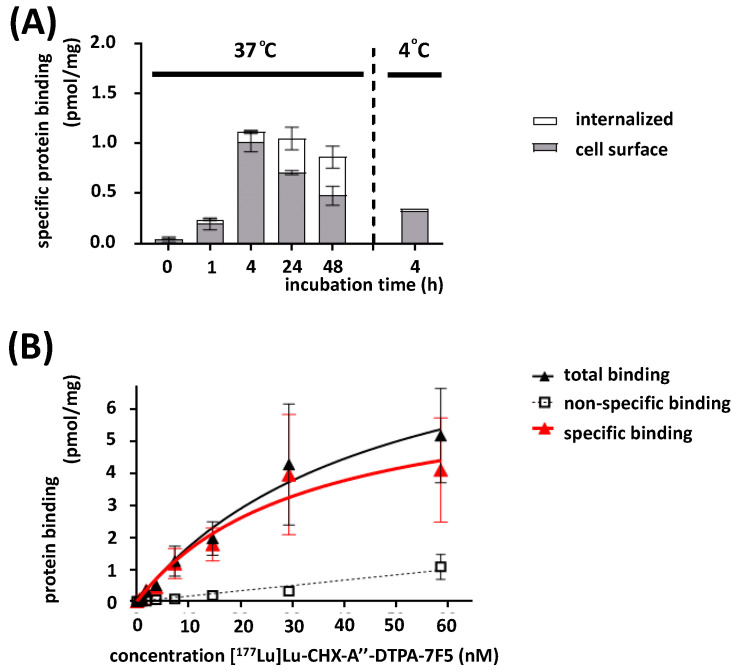
(**A**) Time-dependent PSCA-mediated uptake of [^177^Lu]Lu-CHX-A″-DTPA-7F5 by PC3-P cells. The samples were incubated at 37 °C; at indicated time points, the amount of cell-associated and incorporated radioactivity was determined. (**B**) Representative graph of a saturation binding assay with [^177^Lu]Lu-DTPA-7F5 on intact PC3-P cells. PC3-P cells were incubated for 2 h at 37 °C with increasing [^177^Lu]Lu-CHX-A″-DTPA-7F5 concentrations (n = 3). Nonspecific binding was determined in the presence of 0.5 µM non-labeled anti-PSCA mAb 7F5. Red is the fitted line after subtracting the values of the non-specific binding from those of the total binding.

**Figure 7 ijms-24-09420-f007:**
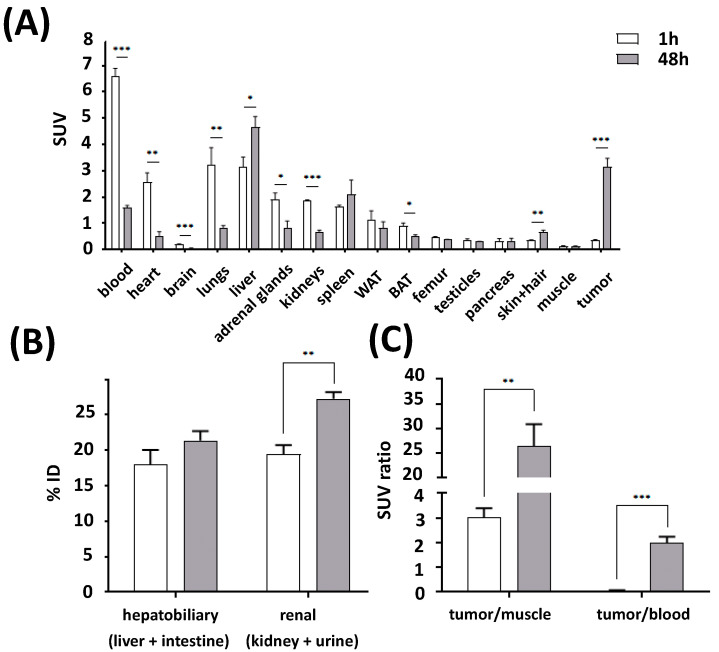
Biodistribution of [^177^Lu]Lu-CHX-A″-DTPA-7F5 in tumor and normal tissues at 1 h and 48 h post injection. [^177^Lu]Lu-CHX-A″-DTPA-7F5 was injected in male PC3-P tumor-bearing mice (n = 4). (**A**) After 1 h and 48 h, SUVs in organs and tissues were determined (mean ± SEM). (**B**) Accumulation of [^177^Lu]Lu-CHX-A″-DTPA-7F5 determined as sum of liver and intestine as well as of kidney and urine. (**C**) SUV ratios of tumor-to-muscle and tumor-to-blood. WAT—white adipose tissue; BAT—brown adipose tissue; *t*-test: * *p* < 0.05; ** *p* < 0.01; *** *p* < 0.001.

**Figure 8 ijms-24-09420-f008:**
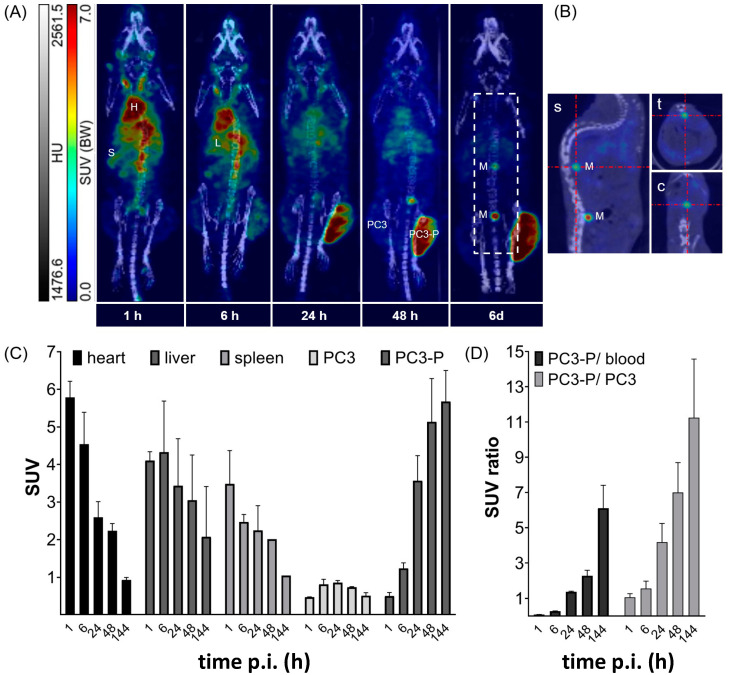
SPECT/CT imaging of [^177^Lu]Lu-CHX-A″-DTPA-7F5 in PC3-P and PC3 tumor-bearing mice between 1 h and 144 h p.i. [^177^Lu]Lu-CHX-A″-DTPA-7F5 was injected in female mice bearing both PC3 (left flank) and PC3-P tumor (right flank) (n = 2). (**A**) Maximum intensity projection (MIP) of the [^177^Lu]Lu-CHX-A″-DTPA-7F5 distribution in a representative tumor xenograft -bearing mouse at 1, 6, 24, 48 and 144 h p.i. H—heart, L—liver, S—spleen, PC3—PC3 tumor, PC3-P—PC3-P tumor, M—potential metastasis. (**B**) Sagital (s), transaxial (t), and coronal (c) sections of two potential metastases near the spinal cord (inside of white box) at 144 h p.i. (**C**) SUV estimated from SPECT images for heart (blood), liver, and spleen, as well as PC-3 and PC3-P tumor at 1, 6, 24, 48, and 144 h p.i. (mean ± SD, n = 2). (**D**) SUV ratios of PC3-P-to-blood and PC3-P-to-PC3 at 1, 6, 24, 48, and 144 h p.i. (mean ± SD, n = 2).

## Data Availability

The data presented in this study are available on request from the corresponding authors.

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
