# Peer review of "Preclinical Characterization of the 177Lu-Labeled Prostate Stem Cell Antigen (PSCA)-Specific Monoclonal Antibody 7F5"

_ijms, 2023, doi:10.3390/ijms24119420_

Round 1

Reviewer 1 Report (New Reviewer)

Herein Striese et al demonstrated the labeling of their PSCA monoclonal antibody 7F5 with 177Lu radionuclide and preclinical characterization in vitro and in tumor xenografted mice model.

Overall, the authors showed quite comprehensive characterizations in terms of labeling and binding in vitro and in vivo. I have only several comments before suggesting the acceptance of this article:

- The authors introduced quite some limitation of current anti PSMA mediated diagnostic, thus proposing for anti PSCA approach. The authors should also mention the limitation associated with anti PSCA

- In terms of stability, the authors demonstrated via reducing non reducing SDS gels. Have the authors characterized aggregation issue of Ab (& conjugated Ab) in presence of serum?

- It is desirable for diagnostic agent to have short circulation lifespan. Have the authors determine the plasma halflife of the proposed Ab?

- The authors showed tumor localization data by subcutaneously injecting 4E6 cells, which is quite large and unlikely to occur in actual setting. Please explain future plan to mediate further translation of the proposed Ab.

- Related to above, the authors utilize the term theranostic instead of diagnostic. Please include some data pertaining therapeutic application of the Ab.

Author Response

Rev 1

Herein Striese et al demonstrated the labeling of their PSCA monoclonal antibody 7F5 with 177Lu radionuclide and preclinical characterization in vitro and in tumor xenografted mice model.

Overall, the authors showed quite comprehensive characterizations in terms of labeling and binding in vitro and in vivo. I have only several comments before suggesting the acceptance of this article:

We are grateful for this overall very positive evaluation of our ms.

- The authors introduced quite some limitation of current anti PSMA mediated diagnostic, thus proposing for anti PSCA approach. The authors should also mention the limitation associated with anti PSCA

We apologize if our writing somehow implicated that PSCA may be a better target than PSMA. Of course ,like PSMA, PSCA is only a tumor associated antigen (TAA) and therefore it is not exclusively expressed in tumor tissues although the expression of PSCA is limited to a few healthy tissues and the level of expression is low. In order to overcome this wrong impression we have completely rewritten this section of the ms and more precisely described in which healthy and tumor tissues PSCA is expressed (/see pages 2 and 3, lanes 90 to 101). 

The text was modified as follows:

Prostate stem cell antigen (PSCA) is such a potential tumor target as it is overexpressed in PCa. Interestingly, PSCA is also overexpressed in a variety of other tumor entities including gallbladder-, urinary bladder-, breast-, renal cell carcinoma, glioma, and pancreatic cancer [25-33]. Moreover, expression of PSCA in PCa increases with the Gleason Score, tumor stage, androgen-independent progression and metastases formation in bone, lymph nodes or liver [32,34,35]. Like PSMA and all other tumor associated antigens (TAAs) PSCA is not exclusively expressed in tumor tissues. Expression of PSCA was also detected at both the mRNA and protein level in healthy tissues. At the mRNA level, PSCA expression could be detetcted in prostate, placenta, kidney and urogenital tissues. At the protein level, expression of PSCA was found in prostate epithelium, epithelial layers of the urinary bladder, neuroendocrine cells of the stomach and colon, collecting ducts of the kidney, trophoblasts of the placenta, and with conflicting reports in pancreas [25-35].

- In terms of stability, the authors demonstrated via reducing non reducing SDS gels. Have the authors characterized aggregation issue of Ab (& conjugated Ab) in presence of serum?

We hope that we correctly understand this critique as the first sentence of the reviewer is scrabled and thus partially unclear. (i) Indeed we performed SDS-PAGE both under reducing and non-reducing conditions. However, the reason of SDS-PAGE was to confirm that the isolated protein is an antibody and it has nothing to do with an analysis of the stability of the protein. As expected of such SDS-PAGE under reducing and non-reducing conditions we estimate a molecular weight of a full size Ig molecule under non-reducing conditions and under reducing conditions we see that Ig molecules is split to the (two) heavy chains and the (two) light chains which become separated and therefore we now find two protein bands according to the molecular weights of a heavy and a light chain.

Obviously, the message of our original text was misleading. We therefore decided to rewrite this portion of the ms as follows (page 3, lanes 119-129)

After purification of the anti-PSCA mAb 7F5 from hybridoma supernatant by protein A affinity chromatography, dialyzed elution fractions were analyzed by SDS-PAGE (Figure 1A, AI) and immunoblotting using an anti-mouse IgG Ab conjugated with alkaline phosphatase ((Figure 1A, AII). SDS-PAGE was performed under non-reducing and reducing conditions (Figure 1A). As expected both conditions resulted in a profile typical for an antibody: Under non-reducing conditions we detected a single protein with a mobility according to a molecular weight of 150 kDa (Figure 1A, AI, left panel, 150 kDa) supporting the purity of the isolated antibody fraction. Under reducing conditions the disulfide bridges between the heavy and light chains of the ab were split resulting in two protein bands according to mobilities of 25 and 50 kDa typical for heavy and light chains of an antibody (Figure 1A, AI, right panel, and AII, 25 and 50 kDa). Moreover, both the nature of the light and the heavy chains could be confirmed by SDS-PAGE/immunoblotting as both protein bands could be detected by an anti-Ig-alkaline phosphatase conjugate (Figure 1A, AII, 25 and 50 kDa).

With respect to the second portion of the question related to the stability of the ab and the formation of aggregates. We have not seen aggregation of the stored antibody including in sera. Moreover, protein aggregates should be seen at the border of the stacking and separation gel and be detectable during SDS-PAGE/immunoblotting. To answer the question of the reviewer we modified the ms as follows (page 3, lanes 129-132):

The isolated antibody fraction does not contain protein aggregates which would be detectable at the border between the stacking and separation gel.

- It is desirable for diagnostic agent to have short circulation lifespan. Have the authors determine the plasma halflife of the proposed Ab?

As mentioned in our introduction section and later shown in the result section (e.g. Figure 1 and 5) we used a full size murine monoclonal antibody in the here presented studies. Like every monoclonal antibody the antibody has a long half life (days to weeks) which is desirable especially for therapeutic purpose. We agree with the reviewer that such a long half live may not be favorable for a rapid diagnostic tool but we are seeking for a theranostic tool that is useful for both diagnosis and therapy. For therapy we need a compound that enriches in a reasonable time and highly specifically at the tumor site where it stays for at least the life span of the radionuclide. Thus, for a theranostic compound one needs to accept a compromise. As the ab enriches excellently at the tumor site in a reasonable time where it should stay for therapy we think that the time required for enrichment is tolerable including for a diagnostic usage. If not, we will construct recombinant derivatives having shorter half lives with sizes between single chain fragments (which might be too short for therapy), bivalent (Fab) like molecules or bivalent minibodies lacking one or the other CH2 domain.

Obviously, we had not sufficiently indicated in the ms that we use a native murine full size monoclonal antibody in our studies. To overcome this limitation we have modified the introduction section (page 3, lane 110) as follows:

In the present study, we conjugated the previously described full size murine anti-PSCA-specific IgG1 mAb 7F5 [40] with the bifunctional chelator p-SCN-CHX-A”-DTPA and radiolabeled the anti-PSCA mAb with the theranostic radionuclide 177Lu.

- The authors showed tumor localization data by subcutaneously injecting 4E6 cells, which is quite large and unlikely to occur in actual setting. Please explain future plan to mediate further translation of the proposed Ab.

As seen in Figure 8 (B), the radiolabelled antibody does not only allow to image the primary injected tumor but even very small metastases. Therefore, we are convinced that the labelled antibody is also useful for detection of tumors having a smaller size.

- Related to above, the authors utilize the term theranostic instead of diagnostic. Please include some data pertaining therapeutic application of the Ab.

We comment to these two requests as follows: As mentioned above we aim to develop the monoclonal antibody as a theranostic tool. Labelling of the murine monoclonal antibody with the theranostic radionuclide 177Lu is a first step in this line. We fully agree with the reviewer that the title of the manuscript may be misleading as we only show so far that the antibody can be labelled with a theranostic radionuclide and enriches in a specific manner at the tumor site where it can be detected by SPECT imaging although in accordance with the excellent tumor uptake of a theranostic radionuclide we expect that we will also see a therapeutic effect. However, to avoid a possible overestimation of our currently available data we have altered the title of the ms as follows:

Preclinical characterization of the 177Lu-labeled prostate stem cell antigen (PSCA)-specific monoclonal antibody 7F5

Reviewer 2 Report (New Reviewer)

In this manuscript, the authors reported a preclinical characterization of a potential immunotheranostic agent for prostate cancer. It discusses the use of a 177Lu-labeled radiolabeled mAb (177Lu-CHX-A’’-DTPA-7F5) to target prostate stem cell antigen (PSCA). They presented the detailed synthetizing process of the mAB and showed the biodistribution and SPECT imaging in vivo studies of the mAB of high specific accumulation of PSCA-positive tumors. This is a very interesting study and the results can help resolve the clinical dilemma that PSMA is not widely expressed in cancer tissue. However, the authors did not seem to mention how effective 177Lu-mAb is in killing cancer cells? Since the word "immunotheranostic agent" is mentioned in the title, this study would be more complete with more descriptions to this point. Some of my other minor comments are also attached below for authors’ reference.

1. PC3-P cells: The authors only stated that this cell line was derived from PC3 and overexpressed PSCA. However, the situation regarding the actual expression levels of PSCA is not fully mentioned. Since this is a man-made cell line and not a real prostate cancer cell, I think the authors have an obligation to disclose the level of PSCA expression, and explain or provide evidence that this level of expression is different from the real world What is the difference between cancer cells.

2. Fig. 6A: In the analysis of internalized mAb, the authors claimed that surface-bound mAb were removed for 5 min with a 4°C cold acid wash buffer. Do the authors have evidence of the effects of stripping? This question is raised because 4°C is a relatively rare stripping condition.

3. Fig. 8: Why are the number of PC3 cells and PC3-P cells injected into naked mice different? Are these two cell lines in nature different? I think the author's description of animal experiment steps is somewhat short, and they should provide more detailed content such as injection locations (left, right), and tumor growth, distribution, and even metastasis. In addition, the resolution of this image is not high enough to read in this version.

Author Response

Rev 2

In this manuscript, the authors reported a preclinical characterization of a potential immunotheranostic agent for prostate cancer. It discusses the use of a 177Lu-labeled radiolabeled mAb (177Lu-CHX-A’’-DTPA-7F5) to target prostate stem cell antigen (PSCA). They presented the detailed synthetizing process of the mAB and showed the biodistribution and SPECT imaging in vivo studies of the mAB of high specific accumulation of PSCA-positive tumors. This is a very interesting study and the results can help resolve the clinical dilemma that PSMA is not widely expressed in cancer tissue. However, the authors did not seem to mention how effective 177Lu-mAb is in killing cancer cells? Since the word "immunotheranostic agent" is mentioned in the title, this study would be more complete with more descriptions to this point. Some of my other minor comments are also attached below for authors’ reference.

First we like to thank for the kind evaluation of our ms. As already mentioned as response to reviewer 1, we agree that the previous title may be misleading as we only show so far that the antibody can be labelled with a theranostic radionuclide and enriches in a specific manner at the tumor site where it can be detected by SPECT imaging. To avoid this possible misunderstanding we have altered the title of the ms as follows:

Preclinical characterization of the 177Lu-labeled prostate stem cell antigen (PSCA)-specific monoclonal antibody 7F5

  1. PC3-P cells: The authors only stated that this cell line was derived from PC3 and overexpressed PSCA. However, the situation regarding the actual expression levels of PSCA is not fully mentioned. Since this is a man-made cell line and not a real prostate cancer cell, I think the authors have an obligation to disclose the level of PSCA expression, and explain or provide evidence that this level of expression is different from the real world What is the difference between cancer cells.

In general, all prostate cancer cell lines downregulate PSCA expression in cell culture including the here used PC3 cell line. The reason is unknown. At least to our knowledge, the only tumor cell line that expresses PSCA in cell culture is the bladder carcinoma cell line RT4. From previous studies we know, that PSCA expression level varies in a wide range in this cell line which seems to affect the growth rate. In order to have a reliable model system for immunotargeting and imaging  constant expression rate of PSCA (and PSMA) we genetically modified the PC3 cells to express PSCA at a constant moderate level of around 25x103 receptors per cell. Compared to other receptors this expression is moderate. For example EGFR is expressed with 1.6x106 rreceptors/cell on A431 cells, 1.6x10receptors/cell on FaDu and 2.2x102 receptors/cell. We like to mention, since construction, our modified PC3 cell lines have been used not only by us but also other colleagues leading to a series of manuscripts (e.g. our own references 38-42). In addition, from immunohistological analyses (e.g. our own reference 30) we know that PSCA expression strongly varies from high to low levels in tumor tissues reaching levels either below or higher compared to our PC3 cell line. So the PSCA expression in our used cell line is not really different from the real world.

As requested we have added the information about the expression level of PSCA in the PC3-P cells (page 12, lanes398 and 399): 

The genetic manipulation leads to a reliable expression at the moderate level of around 25x103 PSCA molecules on the surface per cell.

  1. Fig. 6A: In the analysis of internalized mAb, the authors claimed that surface-bound mAb were removed for 5 min with a 4°C cold acid wash buffer. Do the authors have evidence of the effects of stripping? This question is raised because 4°C is a relatively rare stripping condition.

Stripping of bound radiotracer from cell surface is caused by the acid conditions (0.1 M glycine, 0.15 M NaCl, pH 3.0). This washing step is performed at 4°C to stop all cellular transport processes, especially to stop further uptake of the radiotracer and enzymatic processes including proteolysis. This procedure is a commonly used procedure. It is also compatible to immunoaffinity purification of antigens bound to immunoaffinity columns. 

  1. Fig. 8: Why are the number of PC3 cells and PC3-P cells injected into naked mice different? Are these two cell lines in nature different?

When investigating an radiotracer in a target-expressing tumor (here PC3-P) in comparison to a ‘control’ tumor not expressing the target (here PC3) it is advantageous to compare tumors with approximately the same size to minimize effects due to differences in vascularization, hypoxia, necrosis, and other reasons. As seen in previous studies expression of PSCA seems to slightly accelerate the tumor growth. Based on this experience, we decided to inject an amount of 4×106 PC3 and 3×106 PC3-P to compensate for this difference.

I think the author's description of animal experiment steps is somewhat short, and they should provide more detailed content such as injection locations (left, right), and tumor growth, distribution, and even metastasis. In addition, the resolution of this image is not high enough to read in this version.

We apologize for the poor quality of former Figure 8 (and also some other figures). This must have occurred during formatting of the file. As requested, we improved the quality and resolution of Fig. 8 (and also the other ones) and also added further details of the animal experiment especially to the section 4.11. Animal studies of the Materials and methods section (see page 16, lanes 546 to 552). Overall the text in 4.11. Animal studies was modified as follows (modifications are highlighted in red):

For SPECT imaging, an amount of 4×106 PC3 and 3×106 PC3-P cells was injected subcutaneously in the left and right flank, respectively, of athymic NMRI nude mice (Rj:NMRI-Foxn1nu/nu; Janvier Labs, Le Genest-Saint-Isle, France). For biodistribution, mice were injected with PC3-P cells on their right flank only. Subcutaneous tumors were allowed to grow for about 21 days to reach a tumor size of about 500 mm3. Before radioactivity application the mice weighed 23 ± 1.9 g for biodistribution (n = 8) or 27 ± 1.7 g for SPECT imaging (n = 2).

Reviewer 3 Report (New Reviewer)

The authors present very thorough characterization of an anti-PSCA antibody radioconjugate, including its biophysical properties, in vitro and in vivo activity. The radioconjugate was of high purity and stability, and its specific uptake in the tumors in mice shows favorable biodistribution, so high tumor to background ratio could be revealed using SPECT/CT. The manuscript is clearly written, experiments are well described, chosen methodology suitable and the large dataset well presented (except at points, which I indicated in the list below).  The described molecule extends the armoury of agents that can be used for detection, monitoring and potentially also the therapy of PSCA-positive prostate cancer. My only comment would be that the authors should reveal the isotype (class identity) – of the antibody - I understand that at present, at least a part of it is murine.

Please find below a list of remarks which I hope you will find helpful.

Figure 1C: please show the error bars (it should be easily possible as you have 3 replicates). The presentation could be improved with x-axis in log scale.

Line 135-137: why was an anti-EGFR antibody preferred from using an isotype control? The isotype of the monoclonal 7F5 antibody should be explicitly stated.

Figure 2: please use decimal point and not comma.

Figure 3A: labels of the y-axis are not well legible.

Figure 4A: labels of the right-hand side axes are not well legible, and 4B: labels of the left-side axes are not well-legible. Legend to 4B: please simplify for better legibility: x-axis is once the number of the cells, and once the antigen concentration, and y “the reciprocal of the IRF” and in the graph applied dose/ specific binding.

Lines 202-203: this passage is not too well formed because it is not clear why the SDS-PAGE is indicative of stability? Why were different methods (SDS-PAGE for reduced  and autoradiography of non-reduced material) used?

Figure 6A and B: labels of y-axes are rearranged. 6B: please present the error bars (you indicate triplicate measurements).

Figure 7B. Hepatobiliary pathway lacks significance estimation in the Figure.

Lines 379-381: Please indicate the composition of buffers uniformly with molar concentrations of the constituents and the pH value.

Line 413: PI was also diluted in PBS (I assume?)

Figure S1: A and B, x-axis (cell concentration and inverse cell concentration)-have the same units – please explain.

Figure S2 A: the internalized fractions is not really visible; if this is due to an overlap, please consider presenting it in a separate figure panel. B: text explains certain abbreviations that do not appear in the figure; please amend. Please include the error bars as the text indicates there were 3 replicates. A and B: label text in the y-axis is re-arranged.

Author Response

Rev 3

The authors present very thorough characterization of an anti-PSCA antibody radioconjugate, including its biophysical properties, in vitro and in vivo activity. The radioconjugate was of high purity and stability, and its specific uptake in the tumors in mice shows favorable biodistribution, so high tumor to background ratio could be revealed using SPECT/CT. The manuscript is clearly written, experiments are well described, chosen methodology suitable and the large dataset well presented (except at points, which I indicated in the list below).  The described molecule extends the armoury of agents that can be used for detection, monitoring and potentially also the therapy of PSCA-positive prostate cancer. My only comment would be that the authors should reveal the isotype (class identity) – of the antibody - I understand that at present, at least a part of it is murine. 

Please find below a list of remarks which I hope you will find helpful.

First we also like to thank reviewer 3 for his kind evaluation of our manuscript. We apologize that we had not included to mention that the used antibody is a fully murine monoclonal antibody of an IgG1 isotype.

As also reviewer 1 requested more information about the nature of the monoclonal antibody 7F5, we have included the information about the IgG1 subclass of the 7F5 monoclonal antibody already into the introduction section (page 3, lane 110) as follows:

In the present study, we conjugated the previously described full size murine anti-PSCA-specific IgG1 mAb 7F5 [40] with the bifunctional chelator p-SCN-CHX-A”-DTPA and radiolabeled the anti-PSCA mAb with the theranostic radionuclide 177Lu.

Figure 1C: please show the error bars (it should be easily possible as you have 3 replicates). The presentation could be improved with x-axis in log scale.

We have modified the Figure 1C as requested.

Line 135-137: why was an anti-EGFR antibody preferred from using an isotype control? The isotype of the monoclonal 7F5 antibody should be explicitly stated.

We used the well described anti-EGFR antibody Cetuximabit as an IgG1 isotype control.

Figure 2: please use decimal point and not comma.

We have modified the Figure as requested and further improved the quality of the labelling of the axes.

Figure 3A: labels of the y-axis are not well legible.

We have modified the Figure as requested and thereby improved the quality of the labelling of the axes. The axes are now well legible.

Figure 4A: labels of the right-hand side axes are not well legible, and 4B: labels of the left-side axes are not well-legible. Legend to 4B: please simplify for better legibility: x-axis is once the number of the cells, and once the antigen concentration, and y “the reciprocal of the IRF” and in the graph applied dose/ specific binding. 

We have modified the Figure as requested and thereby improved the quality of the labelling of the axes. The axes are now well legible. In addition, we tried to simplify the figure legend.

Lines 202-203: this passage is not too well formed because it is not clear why the SDS-PAGE is indicative of stability? Why were different methods (SDS-PAGE for reduced  and autoradiography of non-reduced material) used?

To respond to this request we have rewritten this portion of the ms as follows (page 6, lanes 235 to 238) as follows:

To determine its stability [177Lu]Lu-CHX-A’’-DTPA-7F5 was separated via SDS-PAGE two days after radiolabeling (Figure 5A, panel a,b) under reducing (left lanes) and non-reducing conditions ( right lanes). According to silver staining (Figure 5A, panel a)) and autoradiography (Figure 5A, panel b)), the [177Lu]Lu-CHX-A’’-DTPA-7F5 is stable over 2 days.

Figure 6A and B: labels of y-axes are rearranged. 6B: please present the error bars (you indicate triplicate measurements).

We have modified the Figure as requested. Error bars were included.

Figure 7B. Hepatobiliary pathway lacks significance estimation in the Figure.

As there is no significant difference, we have not added a significance estimation.

Lines 379-381: Please indicate the composition of buffers uniformly with molar concentrations of the constituents and the pH value.

We have modified the ms as requested. The buffers are now uniformly given in molar concentrations including pH values.

Line 413: PI was also diluted in PBS (I assume?)

correct

Figure S1: A and B, x-axis (cell concentration and inverse cell concentration)-have the same units – please explain.

We apologize for this obvious mistake and have corrected the dimensions of the x-axis of the Figure S1 respectively (A, cell concentration (106/mL); B, inverse cell concentration (mL/106/mL).

Figure S2 A: the internalized fractions is not really visible; if this is due to an overlap, please consider presenting it in a separate figure panel. B: text explains certain abbreviations that do not appear in the figure; please amend. Please include the error bars as the text indicates there were 3 replicates. A and B: label text in the y-axis is re-arranged.

We have modified the Figure as requested. The error bars are no included. We also removed the abbreviations which are indeed no more existing in the current version of the Figure S2.

Reviewer 4 Report (New Reviewer)

Prostate specific membrane antigen (PSMA) is clinically evaluated to visualize and treat prostate cancer. As with other antigens, however, PSMA is not expressed by all prostate cancer cells, which may reduce the therapeutic efficacy of a radiolabeled ligand targeting it. Striese and colleagues evaluated the membrane protein prostate stem cell antigen (PSCA) as an alternative antigen for the treatment of prostate cancer. The anti-PSCA monoclonal antibody 7F5 has been labeled with the radioisotope lutetium-177 (177Lu), and its ability to bind in vitro to prostate cancer cells and in prostate tumor in an animal model have been determined. The experiments were properly performed and the results support their conclusion.

General comment: Animal models must mimic clinical reality as much as possible in order to properly assess the relevance of a therapeutic modality. The response of cancer cells to radiation is closely associated with immune cells and inflammatory cytokines. An immunocompetent animal model carrying a tumor implanted in the organ where the tumor develops in humans should be favoured. It is therefore strongly suggested to repeat these experiments in animal where the tumor is implanted in the prostate. On the other hand, as mentioned by the authors, immunoradiotherapy or vectorized internal radiotherapy is essentially used in clinic to treat patients with multiple metastases. An animal model with multiple metastases at the same sites as in patients should be used.

Minor comments:

1) Figure 2: Replace “6,8” by “6.8”.

2) Fig. 3 A: To improve the identification of numbers on the X axis, the authors could remove the "0" after the dot. The numbers on the Y axis are not visible.

3) Fig. 4 A. “[177Lu]Lu- -DTPA » on the Z axis must be completely written.

4) PSCA-positive PC3-P and PSCA-negative PC3 tumors that were implanted in the same animal should be better identified on figure 8 A.

5) Fig. 6: Axis titles could be better centralized.

6) Figure 8: The quality of numbers and words needs to be improved to eliminate blurring.

 7) The authors are invited to define the abbreviation “p.i.”.

Author Response

Rev4

Prostate specific membrane antigen (PSMA) is clinically evaluated to visualize and treat prostate cancer. As with other antigens, however, PSMA is not expressed by all prostate cancer cells, which may reduce the therapeutic efficacy of a radiolabeled ligand targeting it. Striese and colleagues evaluated the membrane protein prostate stem cell antigen (PSCA) as an alternative antigen for the treatment of prostate cancer. The anti-PSCA monoclonal antibody 7F5 has been labeled with the radioisotope lutetium-177 (177Lu), and its ability to bind in vitro to prostate cancer cells and in prostate tumor in an animal model have been determined. The experiments were properly performed and the results support their conclusion.

We are also grateful for this evaluation of our manuscript.

General comment: Animal models must mimic clinical reality as much as possible in order to properly assess the relevance of a therapeutic modality. The response of cancer cells to radiation is closely associated with immune cells and inflammatory cytokines. An immunocompetent animal model carrying a tumor implanted in the organ where the tumor develops in humans should be favoured. It is therefore strongly suggested to repeat these experiments in animal where the tumor is implanted in the prostate. On the other hand, as mentioned by the authors, immunoradiotherapy or vectorized internal radiotherapy is essentially used in clinic to treat patients with multiple metastases. An animal model with multiple metastases at the same sites as in patients should be used. 

We agree that animal models must be as close as possible. However, the requested orthotopic model in a fully immunocompetent mouse is not a perfect model. The mouse models we used in the present manuscript has been used by us in a series of previous studies and published manuscripts and were even sufficient and sufficient and acceptable by the authorities to allow us to start clinical phase 1 trials for both a bispecific PSCA-CD3 antibody as well as a UniCAR approach against PSMA. Besides the unnecessary additional animal work for which we would not have and most likely not get the allowance by our authorities to repeat our studies, we believe that our animal data are sufficient as they were also accepted by the other three reviewers.

Minor comments:

1) Figure 2: Replace “6,8” by “6.8”.

2) Fig. 3 A: To improve the identification of numbers on the X axis, the authors could remove the "0" after the dot. The numbers on the Y axis are not visible.

3) Fig. 4 A. “[177Lu]Lu- -DTPA » on the Z axis must be completely written.

4) PSCA-positive PC3-P and PSCA-negative PC3 tumors that were implanted in the same animal should be better identified on figure 8 A.

5) Fig. 6: Axis titles could be better centralized.

6) Figure 8: The quality of numbers and words needs to be improved to eliminate blurring.

 7) The authors are invited to define the abbreviation “p.i.”.

All the requests related to the improvement of the Figures were also made by other reviewers and all Figures were improved accordingly.

This manuscript is a resubmission of an earlier submission. The following is a list of the peer review reports and author responses from that submission.

Round 1

Reviewer 1 Report

The authors describe labeling the prostate stem cell antigen monoclonal antibody 7F5 with Lu-177 and demonstrate targeting to prostate tumors in vivo. The results are encouraging and interesting. However, a control antibody should be included to demonstrate the specificity of the 7F5 antibody. Additional minor revisions are also recommended.

Major Revisions

  1. Figures 6 – 8 – A control antibody should be included to demonstrate the specificity of the [177Lu]Lu-DTPA-7F5 antibody.

Minor Revisions

  1. Figure 2 – Is there any signal for the Triton X-100 treated cells?

  1. Lines 167 – 168 – It is not clear what this sentence means.

  1. Figure 6 – Does bg mean background? This should be described.

  1. Figure 6B – Is the red line labeled as “spec bg” the signal derived from total bg – nonspec bg?

  1. The total number of mice used for all studies should be described.

  1. Lines 278 - 281 – These sentences does not make sense and need to be clarified.

  1. Lines 290 – 292 – This sentence also does not make sense and needs to be clarified.

  1. Line 315 – It is not clear where the 30 nM number comes from. Was this binding affinity determined in one of the figures? If so, this was not clear.

  1. Lines 325 – 330 - These sentences do not make sense and need to be clarified.

  1. Lines 330 – 331 – Low temperature is supposed to “freeze” membranes and stop internalization. Therefore, it is not clear what the authors are trying to say. This should also be clarified.

  1. Lines 368 – 369 – Earlier the authors indicated that 7F5 was unlikely to bind to mouse PSCA. Additionally, in the methods, 7F5 is described as specific for human PSCA. Therefore, the explanation that the spinal localization of the antibody may be due to binding PSCA in the ganglia doesn’t make sense. This should be clarified.

  1. Biodistribution Methods – The type of mouse strain used should be indicated. It is also unclear why urine plus feces activity was used to determine renal and urinary clearance as feces should not clear via that route. This should be explained or modified.

  1. 600 – The authors say “Due to low uptake in non-targeted tissues . . .” However, earlier they mentioned that there should be low uptake in non-targeted tissues as the antibody should not target mouse PSCA. Therefore this conclusion seems inappropriate.

Reviewer 2 Report

Striese and colleagues describe the development and pre-clinical evaluation of a new lutetium labelled moAB targeting PSCA. The study is well described and conducted, and the conclusions appear reasonable. A number of points should be addressed:

  1. Why were PC3-P cells used for in vitro characterisation, and LNCaP-P cells used in the xenografts? PC3 xenografts usually have better take rates and faster growth rates.

  1. In the intro it states: ‘….have already developed metastases at the time of diagnosis, are mostly non-effective.’ There are a number of therapies that are quite effective but are non-curative – suggest rewording.

  1. Reference should be made in the intro to the recently published VISION trial, NEJM 2021

  1. What is the nature of the positive lesions in the spine identified in the imaging studies. Are these metastatic deposits?